# Remote Pilates Training Is Effective in Improving Physical Fitness in Healthy Women: A Randomized Controlled Study

**DOI:** 10.3390/healthcare12070724

**Published:** 2024-03-26

**Authors:** Carine Lazarowitz Zanzuri, Dan Hadas, Yeshayahu Hutzler, Aviva Goral, Sharon Tsuk

**Affiliations:** 1Levinsky-Wingate Academic College, Netanya 4290200, Israel; shayke@l-w.ac.il (Y.H.); avivag@l-w.ac.il (A.G.); sharonts@l-w.ac.il (S.T.); 2Paediatric Cardiology, Shamir (Assaf Harofeh) Medical Center, Zerifin 70300, Israel; danh@shamir.gov.il; 3Israel Sport Center for the Disabled, Ramat-Gan 5253529, Israel

**Keywords:** Zoom, physical activity, sedentary, plank, balance, COVID-19

## Abstract

Despite its positive impact on physical and mental well-being, adults may refrain from performing regular physical activity, due to inadequate time, accessibility, or funds. Yet remote platforms could overcome such obstacles and increase participation. This study evaluated the effectiveness of remote-synchronous group-Pilates classes compared to in-studio classes in healthy sedentary women. In a randomized controlled design, 40 women, aged 20–45, were assigned to a Zoom or studio group-Pilates training. The intervention included twice-weekly 45 min sessions over an eight-week period. Attendance (adherence) was recorded, and the participants completed physical motor tests (plank, curl-up, stork, push-up, and V-sit and reach), Profile of Mood State Surveys, and Nordic Musculoskeletal Pain Questionnaires. Evaluations were performed at baseline, mid-intervention (4 weeks), and post intervention (8 weeks). Adherence to training was high in the Zoom and studio groups (80% and 74%, respectively). Improvements in physical motor tests were seen in both groups following the Pilates interventions, thereby indicating the effectiveness of group-Pilates Zoom training. In conclusion, remote online physical activity such as Pilates offers a good alternative to in-studio trainings, as a means for improving physical fitness and promoting a healthy lifestyle in adults, by offering a more accessible and less timely alternative to in-studio physical activity programs.

## 1. Introduction

Performing physical activity on a regular basis is known to improve physical and mental health and well-being, while increasing longevity rates worldwide [1,2]. As such, the World Health Organization (WHO) recommends performing strengthening activities that involve all major muscle groups, at a moderate-to-high intensity and at least twice a week—in addition to balance training at least once a week [3]. Yet about 31% of adults worldwide have been reported to be insufficiently active [4].

Pilates is a popular modality of physical activity, aimed at improving physical fitness by strengthening muscles and increasing core stability, balance, and flexibility [5,6,7,8]. In addition to its positive effects on physical fitness, Pilates has been shown to reduce resting heart rate (HR) [9] and blood pressure in hypertensive patients [10], to improve mood [11], and decrease musculoskeletal pain [12,13]. Gladwell et al. presented a decrease in self-reported pain following six weeks of Pilates training in people with chronic back pain (using the Oswestry Questionnaire) [14], which is an increasingly prevalent issue among female adults [15]. Yet although most studies focus on back pain, Pilates training was also shown to have positive effects on shoulder pain [16] and knee pain [17].

However, despite the many advantages of physical activity, large numbers of adults are still not active enough to meet the recommended guidelines [18]. Refraining from physical activity participation has been shown to stem from a range of factors, such as inadequate accessibility to facilities, a lack of convenient transportation, insufficient spare time, and high costs [19,20,21,22]. Moreover, studies indicate that women are more likely to be inactive and report insufficient spare time compared to men, mainly due to family caregiving responsibilities [23].

In recent years, new modalities for performing physical activity training from a remote location have emerged and can now be conducted via pre-recorded online videos or synchronous video-conferencing platforms such as Zoom™. Such technology-based remote physical activity training options have several advantages, and could help overcome barriers regarding physical activity participation. The main advantage is their ease of use and accessibility, as they can be performed at any location. A survey conducted during the COVID-19 pandemic showed the feasibility of video training at home, indicated in an 18% increase in new female users in pre-recorded videos or synchronized training, compared to before the pandemic [24]. Moreover, pre-recorded video training that was performed at home was shown to improve physical activity participation in older adults, despite the technological challenges that this population tended to encounter when doing so [25].

An additional advantage of remote physical activity training classes relates to financial savings, which can be seen for both studio owners and participants. For the former, providing online training reduces expenses related to the operating of the training studio or gym (e.g., rental costs, staff wages, and equipment acquisition and maintenance). In turn, this could lower the costs of online classes compared to face-to-face ones, for the benefit of the latter. Moreover, Internet-based training, compared to having to physically travel to a sports center, has been suggested to decrease physical, financial, and mental burdens for patients in general, and for those with a range of musculoskeletal disorders in particular [26]. As such, lowering the cost of physical activity training is an important means for motivating people to participate in organized physical activity training [27].

However, despite these advantages, performing physical activity through a remote online platform may have several disadvantages, such as technological feasibility; the lack of a face-to-face instructor (especially with pre-recorded videos), who can monitor and correct the participants, and clearly demonstrate the exercises; and the lack of social interactions with other participants, which could be a motivating factor for some potential participants [28,29]. Indeed, research indicates that the physical presence of an instructor increases participation in physical activity [28].

Besides the role of the instructors in increasing participation in physical activity, their presence may also positively impact the extent of physical effort exerted during the exercise session. Indeed, a comparison of in-studio training and online video training showed that participants exerted greater physical effort during the former (indicated by the number of sedentary minutes and moderate-to-vigorous minutes per exercise activity) [27]. In addition, participants in pre-recorded exercise training videos exhibited higher HR, energy expenditure, and rate of perceived exertion (RPE), suggesting that participants make greater effort in face-to-face studio sessions [30].

Despite these disadvantages, remote exercise training sessions such as Pilates may have the potential of increasing physical activity participation. It is therefore important to compare remote Pilates training to in-studio classes, while evaluating their effect on improving emotional well-being and physical fitness, as well as additional expected outcomes, such as reduced musculoskeletal pain. The aim of this study, therefore, was to evaluate the effectiveness of group mat-Pilates training conducted synchronously via Zoom, compared to the same training conducted in a studio, in healthy sedentary women.

## 2. Materials and Methods

### 2.1. Participants

The participants included 40 sedentary, non-physically active, healthy women that were recruited through advertisements on social media. Participant age range was 20–45 years, the average age = 30.45 ± 5.8 years; and average body mass index [BMI] = 22.8 ± 5.8. Most participants were single (80%), had no children (85%), and held full-time employment (87%). Two inclusion criteria were implemented in this study: (1) women who had not participated in physical activity on a regular basis over the past six months; and (2) women who did not take medication on a regular basis (excluding birth control pills). Four exclusion criteria were also implemented in this studying, including (1) women who were pregnant; (2) women who reported having a chronic disease or injury; (3) women who planned to change their diet or physical activity routine during the study intervention period; and (4) women who were not approved by the physician during the evaluation session to take part in this study.

### 2.2. Procedure

For this study, a randomized controlled intervention program was applied, comprising twice-weekly group-Pilates training over an eight-week period. The research group performed their Pilates training via the remote Zoom platform (San Jose, CA, USA), while the control group performed the same type of physical activity in a studio. A schematic presentation of study procedure is presented in Figure 1.

A randomized controlled design (stratified randomization) was conducted and counter-balanced for physical ability, by rating the plank test results at the baseline [31]. In this manner, the participant with the highest plank test result was randomly assigned to one of the two groups, and the participant with the second highest result was then assigned to the other group. This was then conducted for the third and fourth highest ranking participants on this test, and so forth.

In addition, the participants were assessed for motor ability, mood, and pain at four different time points: twice prior to the intervention, once four weeks into the intervention, and eight weeks into the intervention. The period between the first and second assessments prior to the intervention served as a baseline non-intervention control period and for evaluating the testing effect.

During the first evaluation session, the participants received an explanation about the study by the physician, and were asked to sign a written informed consent form. They then underwent a physical examination, conducted by a physician, who approved their participation in the study. They also performed five physical motor tests and completed two questionnaires: Profile of Mood State (POMS) and Nordic Musculoskeletal Pain (NMQ). The participants were asked back for a second baseline evaluation about two weeks later.

### 2.3. Intervention

The intervention program included mat-Pilates group training that was conducted via Zoom or in a studio, both types conducted by two female, certified, and experienced Pilates instructors. Employing two instructors in this study, rather than just one, was beneficial as it eliminated the impact of a specific instructor’s personality on the physical activity participation and outcomes, as both instructors taught in both groups. Finally, it is important to note that the Pilates instructors were blinded to the aim of the study, and were only informed that the study strove to evaluate the impact of Pilates exercises on physical fitness.

### 2.4. The Pilates Exercise Program

Both Pilates training programs (Zoom and in-studio) consisted of twice-weekly 45 min sessions and followed the same exercise routines. While basic levels of Pilates were performed during the first four weeks of the intervention, moderate levels of Pilates were performed during the final four weeks, as a means for exposing the participants to increasing levels of difficulty. Both instructors received the same guidelines regarding the progression of the Pilates training, and were asked to pay attention to the capabilities of each individual participant. This enabled the instructors to modify the intensity of a given exercise to suit a participant’s capabilities, offering easier or more difficult variations when needed, in turn enabling the participants to progress at their own pace. Finally, the instructors were asked to place an emphasis on balance, flexibility, and core muscle exercises, while continually providing feedback and motivating the participants throughout the program.

### 2.5. The Zoom Exercise Group

For the synchronous Zoom sessions, the participants performed the exercises from the comfort of their homes via their personal computer; throughout the session, the instructor could communicate with them. Each participant could see herself and the instructor on their computer screen; if they so desired, they could also observe the other participants. Using the Zoom platform also enabled the instructor to see the name of each participant throughout the session. The instructors watched the participants on a 40″ television screen, and were asked to pay attention to each individual participant, while providing both general and personal feedback, as per the recommendations of Turolla et al. [32].

### 2.6. The Studio Exercise Group

The face-to-face sessions were conducted in a spacious fitness studio, with one of the participating instructors. The instructors received the same guidelines for both Zoom and in-studio training sessions.

#### 2.6.1. Measurements

Personal information about each participant was gathered via a questionnaire that included 11 items relating to the participant’s socio-demographic characteristics and health status. The participants were also asked about their current participation in physical activity, to ensure that the participants had not conducted physical activity on a regular basis for the past six months. The questionnaire was completed through a phone call with one of the researchers, prior to the intervention program, at a date and time of the participants’ convenience.

Adherence to training: Attendance for each participant was documented by the instructors at the beginning of each session, and then calculated as a percentage of all 16 sessions (i.e., the number of sessions attended/the total number of sessions × 100).

Anthropometric measurements: Three physical factors were measured for each participant: (1) height was measured using a roll-up measuring tape with a wall attachment (Seca, model 206), prior to the intervention; (2) weight was measured using digital scales (Life, weight range: 2.5–180 kg) at all time points; and (3) resting HR was measured using the Polar FT7 watch, after sitting on a chair for five minutes, and at the same time point as the weight assessments.

Motor tests: Five motor tests were conducted (plank, curl-up, stork, push-up, and V-sit and reach tests) at all time points. (1) The plank test is used to measure core muscular strength and endurance [33]. In this study, the participants were asked to maintain the plank position on a mat on the floor for as long as possible. The participants were instructed to hold their body in a straight, push-up-like position, hips elevated, with the body’s weight being supported by their elbows, forearms, and toes. The duration of maintaining this plank position was timed using a stopwatch and then documented. (2) The curl-up test is used to measure abdominal muscle strength [34]. In this study, the participants were asked to lie on a mat on the floor in a supine position, knees bent, feet on the floor, arms straight alongside the body, and hands pronated with straight fingers. The participants were then asked to curl up and down as many times as possible throughout a one-minute duration, while maintaining their heels and fingertips on the mat at all times. The number of curl-up repetitions was recorded and documented by the instructor. (3) The stork test is used to measure static balance and core stability [35]. In this study, the participants were asked to stand on one (supporting) leg, with the foot of the other leg pressed against the supporting knee, hands on hips. At the “go” signal, the participants were instructed to raise their non-supporting leg off the floor and stand in the required position for as long as possible. The test ended when their heel touched the ground, or when movement was seen in the supporting foot, and the duration was recorded in seconds. (4) The push-up test is used to measure upper-body muscular endurance strength [36]. In this study, the participants were asked to perform a modified push-up test (with their knees on the floor) as many times as possible during a one-minute duration. The number of repetitions was recorded and documented by the instructor. Finally, (5) the V-sit and reach test is used to measure posterior chain flexibility and trunk mobility (hamstring and back) [37]. In this study, the participants sat on the floor with their legs in a V-shaped position of hip width (approximately). A piece of tape was placed on the floor between their knees (23 cm width), and the participants were asked to slowly bend forward as far as possible, to touch the tape with their hands, and then remain in that position for approximately 2 s. The reached distance was measured in cm, and the best attempt out of three was recorded and documented by the instructor.

Profile of Mood State Questionnaire (POMS): This questionnaire measures a person’s mood state on a given day [38]. The questionnaire was comprised of 28 items rated on a Likert-like scale, ranging from 0 (not at all) to 5 (very true). The items were divided into the following five sub-domains: (1) vigor (items 5, 7, 9, 17, 24, 26, and 27); (2) fatigue (items 3, 12, 18, 23, and 28); (3) tension (items 1, 10, and 19); (4) depression (items 4, 8, 11, 15, 16, 21, and 22); and (5) anger (items 2, 6, 13, 14, 20, and 25). To calculate the score, the ratings for each domain were summed up. In the vigor domain, higher scores indicate a better mood, while for all other domains, higher scores indicate a worse mood. This questionnaire was completed by the participants at all time points.

Nordic Musculoskeletal Questionnaire (NMQ): This questionnaire assesses musculoskeletal pain symptoms in different body regions, based on 28 multiple-choice questions that comprise two well-differentiated parts. The general section addresses symptoms in nine parts of the body (neck, shoulders, elbows, wrists/hands, upper back, lower back, hip/thighs, knees, and ankles/feet) over the previous 12 months/7 days [39]. The second part refers to symptoms in the lower back (0 = no pain; 1 = pain). The questionnaire was completed at all time points. An extended version of the NMQ has been validated in Hebrew, and is frequently used for practical evaluations in physiotherapy treatment [40]. After analyzing the NMQ, and since no indication was seen for a dominant anatomical location of pain among the participants, analyses were performed regardless of pain location.

Feeling scale (FS): This scale measures affective responses, pleasure, and satisfaction after a training session. At the end of each session, both groups of participants were asked to access a dedicated mobile-phone application, to rate their current feelings on an 11-point scale, ranging from +5 (very good), through 0 (neutral), to −5 (very bad) [41]. This scale enabled the monitoring of the participants’ satisfaction after each training session, through the entire intervention period. The average score for each group was calculated per session.

Delayed onset muscle soreness (DOMS): The participants were also asked, 24 h after each session, if they had any muscle pain or soreness following the previous day’s Pilates session (yes/no) [42]. This question was sent to each participant via a dedicated mobile-phone application.

#### 2.6.2. Statistical Analysis

The following post hoc sample size calculation was conducted to ensure the power of the sample. The sample size was calculated using G*POWER (software 3.1.9.4) based on a partial eta squared of 0.06, an effect size of 0.252, an α of 0.05, and a power of 0.8. Accordingly, the sample size was determined to be 70. However, due to logistical difficulties due to COVID-19 restrictions, the final number of participants was 40, with the calculated power being 0.52. All numeric variables were assessed for normality using Skewness and Kurtosis. The participants’ BMI, HR rest, and physical motor abilities were evaluated at the two baseline time points, to evaluate learning/testing effects using independent sample t-tests. The variables that were measured at the second baseline time point were used as a pre-intervention baseline for further analysis.

The five motor tests, POMS, and FS were analyzed using Bayesian statistics of repeated-measures analysis of variance (ANOVA). The hypothesis of this study was that similar motor improvements will be seen in all participants, both those who perform in-studio Pilates and those who perform Pilates via Zoom. As such, traditional analysis might not be suitable for a ‘no difference’ hypothesis [43].

The Bayesian analysis consisted of three models: (1) group difference; (2) pre–post difference (intervention); and (3) interaction of group and intervention. The categorical variables, DOMS and NMQ, were analyzed using Bayesian contingency statistics. Each of these three models were compared to a null model (BF_10_), which indicated that there was no difference between the compared values. Models yielding a BF_10_ > −1 were considered as having an effect. In addition, adherence to training was compared between the two groups using independent sample *t*-tests. It should be noted that two participants were not included in the final sample and in the statistical analyses: one participant withdrew from the Zoom intervention during week 6, and one participant from the in-studio group was not present during some of the assessments. Finally, effect sizes of the motor tests were calculated using Cohen’s d, with effect sizes being classified as small (d = 0.2), medium (d = 0.5), and large (d ≥ 0.8) within pre–post differences for the baseline vs. the post-intervention time point.

## 3. Results

At the baseline time point, no differences were seen in the participants’ characteristics between the Zoom group and the in-studio one (Table 1). Though participants were only counterbalanced between the groups based on their plank ability, the results of the other four motor tests were also similar between the two groups.

When examining the effectiveness of the Zoom intervention program compared to the parallel in-studio one, similar levels of participation were seen, with no significant differences between the two groups (80% and 74%, respectively; *p* = 0.26). Moreover, no significant differences were seen between the two groups in their reported levels of satisfaction via the FS (the group model of Bayesian statistics was 0.82). The mean scores for the Zoom group and for the in-studio group were 3.0 and 3.6, respectively—both representing good scores. In addition, when comparing the motor test results of the two groups (Table 2a), improved motor abilities were seen in both groups following the intervention, with no significant differences between the groups (Table 2b).

Finally, when evaluating the effectiveness of the Zoom training in reducing musculoskeletal pain, a small decrease was seen in self-reported pain in the Zoom group (intervention model = 1.8). Yet no significant differences were seen between the groups in this regard (group model of Bayesian statistics = 0.38). Moreover, when evaluating the physical soreness after each session, reported via the DOMS Questionnaire, no significant differences were seen between the two groups (the group model of Bayesian statistics was 0.03, although this decreased over time, as expected; the studio intervention model was 2.2 and the Zoom intervention model was 5.1; both models are above the null model). Finally, when evaluating the participants’ mood, reported via the POMS Questionnaire, no significant differences were seen within each group (intervention model of Bayesian statistics = 0.85) or between the two groups (group model of Bayesian statistics = 0.34).

## 4. Discussion

The aim of this study was to evaluate the effectiveness of mat-Pilates training via the online synchronous Zoom platform, compared to similar in-studio training. Our findings highlight the feasibility of the Zoom training, as seen through the high participant adherence rate. Adherence, a basic requirement for participating in an exercise program [4], is well documented as having a positive impact on health [3]. The rates of adherence in both groups are in line with results presented in a recent systemic review [44]. While adherence to remote physical training has not yet been adequately documented and evaluated, Schwartz et al. (2021) found very high adherence to physical resistance training that was conducted via Zoom during COVID-19 lockdowns [45].

Interpersonal relationships, which are considered an important factor for increasing adherence to physical activity participation, are missing in remote exercise programs [46]. Yet high adherence was seen in both groups in the current study; as such, it seems that a lack of face-to-face interpersonal interactions did not hinder adherence in the Zoom group. The high adherence in the Zoom training might be related to reasons other than meeting people. Adherence to physical activity has also been shown to be highly dependent on self-efficacy and motivation to exercise [47]. Although these factors were not directly evaluated in our study, both groups of participants reported feeling good after most exercise sessions, which may indicate their satisfaction with the exercise trainings. Moreover, self-efficacy and motivation to participate in physical activity is also influenced by the instructor–participant relationship, and by the feedback provided by the instructor during the session [48]. Yet in remote exercise modalities, providing personal or general feedback is more challenging. While instructor feedback has been shown to play an important role in participant satisfaction [46], the lack of face-to-face instructor in the Zoom group did not appear to have a negative impact on participants’ motivation to participate in the Zoom training sessions in this study.

The effectiveness of physical activity programs should mainly be evaluated through improvements in the physical ability of participants. In this study, the Pilates training in both groups improved muscle strength, flexibility, and balance—as is expected from Pilates training [49,50,51]. In most motor tests that were evaluated in this study (four out of five), the effect sizes were similar between the Zoom and in-studio groups, except for the stork balance test. The increase in muscle strength after the Pilates training was demonstrated in improvements in the curl-up and push-up tests. The large effect size of curl up tests resembles the study of Rayes et al., who evaluated eight weeks of Pilates studio training for obese men and women [49]. In addition, Suner-Keklik et al. showed that six weeks of synchronized on-line Zoom Pilates training improved curl up test results in young healthy women, and the effect size was also large [51]. The effect sizes of the push-up and V-sit and reach tests in our study, which ranged from medium to large, were also compatible with other Pilates studies [50,52].

In contrast to the large effect on muscle strength, the Pilates training had a small effect on core body stability, measured by the plank test. This finding differs from previous studies, that show improvement, for example, after six weeks of Pilates Zoom training three times per week [51].

It was of concern that performing Pilates training via a remote platform may lead participants to exert less physical effort than during in-studio training. This was seen, for example, in female college students, who were shown to elicit less effort during remote exercise compared to in-studio training [2]. While we did not measure the levels of exerted effort in this study, it was expected that even minimal effort on the part of the participants (who were sedentary novice Pilates trainees) would cause some degree of muscle ache. This expectation is based on findings whereby an acute bout of one single Pilates session caused muscle damage in women who had not participated in physical activity on a regular basis over the previous six months [53]. In this study, the level of effort was indirectly evaluated by the participants’ DOMS reports, with similar levels of muscle soreness actually reported by both groups of participants. As such, we can assume that a reasonable degree of effort was exerted in the Zoom group.

In addition, Pilates is renowned for its ability to attenuate musculoskeletal pain, mainly in the lower back [13,54,55]. In this study, an overall small reduction in pain was seen at the end of the eight-week intervention in both groups. This reduction in pain is similar to that presented by Askari et al., whereby women who performed Pilates three times a week over a six-week period reported reduced lower back pain [56].

The effectiveness of physical activity on a regular basis may affect physical parameters such as BMI and resting HR. Nevertheless, no such changes were seen in either group in this study after the eight weeks of Pilates training. Similar to our study, a lack of improvements in weight or BMI was reported in other Pilates training programs, such as in obese women, when comparing online and in-studio Pilates programs [57] and in sedentary middle-age women [58]. If participants in Pilates classes are interested in losing weight, their instructors should encourage them to also participate in aerobic training, in addition to Pilates.

No changes were seen in the participants’ resting HR in either group following the eight-week intervention. Suna et al. found that eight weeks of Pilates training, three times a week, led to a small reduction in resting HR in sedentary women [9]. The small-to-no change in the participants’ BMI and resting HR could indicate that Pilates is not a form of physical training that could impact such physiological parameters—whether conducted in the studio or remotely via Zoom. Finally, while physical improvements may be expected among young, healthy, and untrained women following either in-studio or remote Zoom sessions, it may be of interest to also explore additional interventions, such as pre-recorded videos or app-based training, that do not require the supervision of an instructor [59].

## 5. Limitations

Despite its contribution to the literature, this study has several limitations that should be addressed. First, a relatively small sample size was employed, due to the COVID-19 pandemic that greatly restricted social gatherings. As such, any generalization of these findings should be undertaken with caution. However, such a sample size is similar to that seen in an earlier comparative study [60]. The pandemic also limited the intervention period, which resulted in 16 sessions over an eight-week period. Additionally, this study focused on young, sedentary, healthy women. Future studies could therefore benefit from including additional populations, such as older or active women. Moreover, some of the data analyzed in this study were gathered using the self-reporting method. Finally, no direct assessments of the exerted effort during the training sessions were conducted. Such assessments could be beneficial in future research.

## 6. Conclusions

Remote mat-Pilates group training via the Zoom platform offers a feasible and effective alternative to in-studio mat-Pilates sessions in healthy sedentary women. This option could help increase participation in regular physical activities, for the benefit of the participants, the healthcare systems, and society as a whole.

## Figures and Tables

**Figure 1 healthcare-12-00724-f001:**
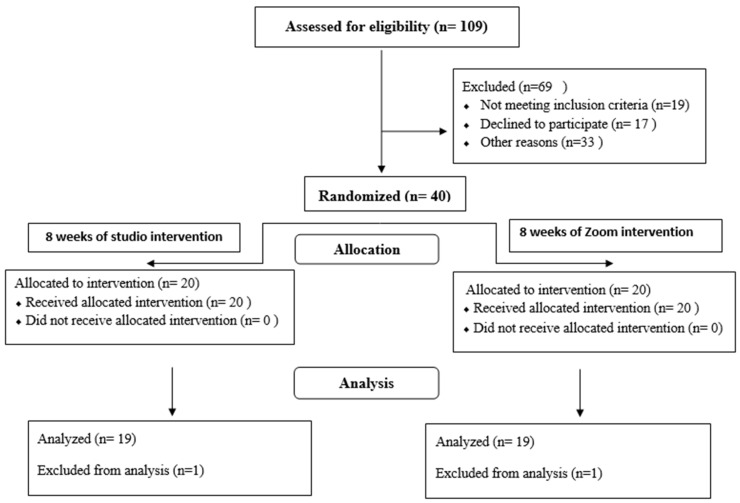
Consort chart of study procedure.

**Table 1 healthcare-12-00724-t001:** Baseline characteristics of participants presented as means (SD) or percentages.

	Studio Group (n = 19)	Zoom Group (n = 19)	*p* Value
BMI	22.7 (3.3)	22.9 (3.3)	0.89
HR rest (beat/min)	78 (11.1)	75.5 (6.4)	0.27
Curl-up test (rep/min)	34.2 (9.7)	34 (6.1)	0.86
Push-up test (rep/min)	24.9 (9.2)	23.3 (8.1)	0.58
V-sit and reach (cm)	47.1 (11.4)	53.2 (10.3)	0.86
Plank test (s)	67.1 (30.1)	66.2 (25.7)	0.93
Stork test (s)	2.3 (1.2)	2.9 (1.4)	0.36
Working status (full-time job)	46%	54%	0.34

*p* values indicate the differences between the groups.

**Table 2 healthcare-12-00724-t002:** (**a**) Mean (SD) motor test scores at the baseline, 4 weeks, and 8 weeks into the intervention. (**b**) Bayesian model values of motor tests for explaining differences between the groups and differences between the baseline, 4 weeks, and 8 weeks into the intervention.

(a)								
	Studio Group (n = 19)				Zoom Group (n = 19)			
Test	Baseline	Middle	End	Effect Size	Baseline	Middle	End	Effect Size
Curl-up test (rep/min)	34.6 (9.8)	37.9 (6.4)	41.8 (8.4)	0.8	33.6 (6.2)	40.3 (8.2)	43.8 (6.7)	1.6
Push-up test (rep/min)	25.9 (8.7)	30.4 (7.2)	34.4 (8.40)	1	23.0 (8.2)	29.5 (10.1)	30.1 (11.4)	0.7
V-sit and Reach test (cm)	47.3 (8.7)	53.4 (7.2)	53.9 (8.4)	0.8	54.1 (8.2)	59.0 (10.1)	60.6 (11.4)	0.6
Plank test (s)	66.1 (30.2)	73.0 (23.6)	78.7 (31.5)	0.4	65.1 (26)	71.5 (28.6)	70.6 (27.4)	0.2
Stork test (s)	2.3 (1.2)	3.7 (1.7)	3.5 (1.7)	0.8	2.9 (1.44)	3.9 (3.10)	3.3 (2.3)	0.2
**(b)**								
		**Bayesian Factor_10_**						
		**Group Model**		**Intervention Model**		**Interaction Model**		
Curl-up test		0.3		**2.10 × 10^6^**		2.60 × 10^5^		
Push-up test		0.5		**2.35 × 10^6^**		4.41 × 10^5^		
Stork test		0.4		**1.18 × 10^1^**		1.05 × 10^0^		
V-sit and reach test		1.9		**2.37 × 10^9^**		8.64 × 10^8^		
Plank test		0.5		**3.35 × 10^0^**		4.70 × 10^−1^		

Baseline = prior to intervention; Middle = 4 weeks into the intervention; End = 8 weeks into the intervention. Size effects were calculated with Cohen’s d standards, between baseline and end. Bold markings indicate the highest model for explaining the effects.

## Data Availability

The data presented in this study are available upon request from the first or corresponding author.

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
