# Peer review of "Remote Pilates Training Is Effective in Improving Physical Fitness in Healthy Women: A Randomized Controlled Study"

_healthcare, 2024, doi:10.3390/healthcare12070724_

Round 1

Reviewer 1 Report

Comments and Suggestions for Authors

Dear author

I am interested in this research that has shown the effectiveness of the remote Pilates training. I think that there are minor revision things as follows;

1)     Please write the reason why one participant in each group was excluded in analysis stage.

2)     In intervention section, please write one-exercise time in remote and in-studio groups.

From reviewer.

Author Response

We would like to thank the reviewer for their insightful questions and input. Please find following our response to each comment, marked in bold.

1) Please write the reason why one participant in each group was excluded in analysis stage.

Thank you for pointing out this oversight. In section 2.6.2 Statistical Analysis, it now reads: “It should be noted that two participants were not included in the final sample and in the statistical analyses: one participant withdrew from the Zoom intervention during week 6, and one participant from the in-studio group was not present during some of the assessments.”

2) In intervention section, please write one-exercise time in remote and in-studio groups.

Again, we apologize for this oversight. This does appear in the Abstract and we have now also added the following sentence in Section 2.4 The Pilates Exercise Program: “Both Pilates training programs (Zoom and in-studio) consisted of twice-weekly 45-minute sessions and followed the same exercise routines.”  (Lines 148–149).

Reviewer 2 Report

Comments and Suggestions for Authors

Dear authors, 

The manuscript is well-writen and the theme is interesting. The following recommendations were done to improve the draft's quality. Also, please see the attached file for minor addressed issues. Regards.

- Insert a post hoc sample size calculation to ensure the actual power of the sample.

- Insert in the participants' section the Ethical approval number and the clinical trial registration.

- Whe did you performed a per protocol analysis instead of the intention-to-treat? 

- Insert references to justify the type of participants' asignment at the randomization procedure.

- Were the questionnaires validated? Please, insert at least a single psychometric study at the references.

- Please, insert the qualitative classification used for Cohen d's values.

- In the results section, please make sure the factors are well stated. I could not notice the values for between and within-factors. My suggestion is to perform a factorial ANOVA with rep measures and a post hoc test.

- Review the limitation section. I believe you may clarify other important limitations of your study. 

I hope the above queries may improve the final quality of your paper.

Regards.

Comments on the Quality of English Language

The manuscript would benefit of a Native English review.

Author Response

We would like to thank the reviewer for their insightful questions and input. Please find following our response to each comment, marked in bold.

1) Insert a post hoc sample size calculation to ensure the actual power of the sample

The following sentences have now been added to the revised manuscript: “The following post hoc sample size calculation was conducted to ensure the power of the sample. The Sample size was calculated by G* POWER (software 3.1.9.4) based on partial eta squared of 0.06, effect size – 0.252, α -0.05, Power – 0.8. Accordingly, the sample size was determined to be 70. However, due to logistical difficulties dur to COVID-19 restrictions, the final number of participants was 40, with the calculated power being 0.52.” (Lines 252-255).

2) Insert in the participants' section the Ethical approval number and the clinical trial registration.

As per the journal’s guidelines, this information appears at the end of the manuscript as follows: Institutional Review Board Statement: "The study was conducted in line with the guidelines of the Declaration of Helsinki, and was approved by the Ethics Committee of the Shamir (Assaf Harofeh) Medical Center, Israel. (Protocol code 0121-21–ASF on April 18, 2021)". We could move this to the Participants section if needed.

3) Why did you performed a per protocol analysis instead of the intention-to-treat? 

We have added the following sentences to of the revised manuscript to the statistical analysis section: "One participant from the ZG withdrew from the program at week 6, and another participant from the SG missed time 2 tests and therefore were not included in the final sample and statistical analyses".

4) Insert references to justify the type of participants' assignment at the randomization procedure.

We have added the following reference for the randomized procedure employed in this study:

Suresh K. An overview of randomization techniques: An unbiased assessment of outcome in clinical research. J Hum Reprod Sci. 2011 Jan;4(1):8-11. (Lines 120–121).

5) Were the questionnaires validated? Please, insert at least a single psychometric study at the references.

The POMS questionnaire in Hebrew had been previously validated, as per reference no.38. (Line 221). The extended version of the NMQ in Hebrew had also been previously validated, as per reference nos. 39 and 40 (Lines 233 and 236).

6) Please, insert the qualitative classification used for Cohen d's values.

You are correct. Thank you for pointing out this oversight. The following wording has now been added to the revised manuscript, Section 2.6.2 Statistical Analysis: “Finally, effect sizes of the motor tests were calculated using Cohen’s d, with effect sizes being classified as small (d = 0.2), medium (d =0.5), and large (d≥0.8) within pre-post differences for the baseline vs. the post-intervention.” (Lines 275–278).

7) In the results section, please make sure the factors are well stated. I could not notice the values for between and within-factors. My suggestion is to perform a factorial ANOVA with rep measures and a post hoc test.

We have added the following sentences to the revised manuscript, the 2.6.2 statistical analysis section: “The hypothesis of this study was that similar motor improvements will be seen in all participants, both those who perform in-studio Pilates and those who perform Pilates via Zoom. As such, traditional analysis might not be suitable for a 'no difference' hypothesis (43). (Lines 262–265).

8) Review the limitation section. I believe you may clarify other important limitations of your study. 

Thank you for this important comment. We have clarified additional limitations in the revised manuscript as follows: “Despite its contribution to the literature, this study has several limitations that should be addressed. First, a relatively small sample size was employed, due to the Covid-19 pandemic that greatly restricted social gatherings. As such, generalization of these findings should be made with caution. However, such a sample size is similar to that seen in an earlier comparative study (59). The pandemic also limited the intervention period, which resulted in 16 sessions over an eight-week period. Additionally, this study focused on young, sedentary, healthy women. Future studies could therefore benefit from including additional populations, such as older or active women. Moreover, some of the data analyzed in this study was gathered using the self-reporting method. Finally, no direct assessments of the exerted effort during the training sessions were conducted. Such assessments could be beneficial in future research.” (Lines 391–401).

Reviewer 3 Report

Comments and Suggestions for Authors

The article raises an important issue: increasing the participation of adults in physical activity. This is possible by expanding the offer of training methods. An interesting method adapted to current conditions may be training from a remote location.

The authors' intention was to evaluate the effectiveness of group mat-Pilates training conducted synchronously via Zoom, compared to the same training conducted in a studio, in healthy sedentary women. The entire experiment was very well prepared and carried out, and the description of the research procedure and the results obtained raise no objections. Summarizing their study, the authors concluded that remote mat-Pilates group training via the Zoom platform offers a feasible and effective alternative to in-studio mat-Pilates sessions in healthy sedentary women. In my opinion, this result was predictable without such detailed and expensive research. In the experiment described in the article, both training groups consisted of women with a high level of motivation, which they declared when participating in the study. The effect of training depends primarily on participation in physical activity and correct performance of given exercises, which was achieved by ensuring the constant presence of the instructor in both groups. Therefore, women exercising at home had a more comfortable situation because they did not have to waste time traveling to a sports center. In my opinion, it would be much more useful to compare the effect of training in another group of women exercising at home without the supervision of an instructor and using pre-recorded videos or synchronized training. I propose to expand the discussion by considering this problem.

Author Response

We would like to thank the reviewer for reading our manuscript and for offering important insights and comments.

The authors' intention was to evaluate the effectiveness of group mat-Pilates training conducted synchronously via Zoom, compared to the same training conducted in a studio, in healthy sedentary women. The entire experiment was very well prepared and carried out, and the description of the research procedure and the results obtained raise no objections. Summarizing their study, the authors concluded that remote mat-Pilates group training via the Zoom platform offers a feasible and effective alternative to in-studio mat-Pilates sessions in healthy sedentary women. In my opinion, this result was predictable without such detailed and expensive research. In the experiment described in the article, both training groups consisted of women with a high level of motivation, which they declared when participating in the study. The effect of training depends primarily on participation in physical activity and correct performance of given exercises, which was achieved by ensuring the constant presence of the instructor in both groups. Therefore, women exercising at home had a more comfortable situation because they did not have to waste time traveling to a sports center. In my opinion, it would be much more useful to compare the effect of training in another group of women exercising at home without the supervision of an instructor and using pre-recorded videos or synchronized training. I propose to expand the discussion by considering this problem.

Thank you for this proposal. We appreciate your input and have added the following sentences to the Discussion chapter in the revised manuscript: “Finally, while physical improvements may be expected among young, healthy, and untrained women following either in-studio or remote Zoom sessions, it may be of interest to also explore additional interventions, such as pre-recorded videos or app-based training that do not require the supervision of an instructor.” (Lines 386–389).

Reviewer 4 Report

Comments and Suggestions for Authors

Why only eight weeks, not more (12 weeks), and the abandonment after?

Assessment of physiologic parameters? Such HR? Why only questionnaires?

It is missing a control group to compare the results with the other two.

Author Response

We would like to thank the reviewer for their important and insightful questions and input.

1) Why only eight weeks, not more (12 weeks), and the abandonment after?

Due to the restrictions of the Covid-19 pandemic, we were unable to continue the intervention for a longer period. We have added this to section 5 Limitation.

2) Assessment of physiologic parameters? Such HR? Why only questionnaires?

Please note that a number of physiological parameters were assessed in this study. For example, resting HR (Lines 187, 257, 374, 383–386), as well physical motor tests (plank, curl-up, stork, push-up, and V-sit and reach).

3) It is missing a control group to compare the results with the other two.

In this study, two groups were compared – an in-studio Pilates group and a Zoom Pilates group. The aim was to compare the outcomes of the two. A third (control) group could have added more information, as you suggest. Yet this was not possible due to the Covid-19 restrictions that greatly limited the sample size and intervention length. However, Prior to the intervention, the participants underwent two evaluations, with at least two weeks between them. This served as the baseline for all further analyses.